# Psychache, Hopelessness, and Suicidal Ideation and Behaviors: A Cross-Sectional Study from China

**DOI:** 10.3390/ijerph21070885

**Published:** 2024-07-08

**Authors:** Ching Sin Siau, E. David Klonsky, Kairi Kõlves, Jenny Mei Yiu Huen, Caryn Mei Hsien Chan, Muhamad Nur Fariduddin, Norhayati Ibrahim, Yee Kee Tan, Cunxian Jia, Jie Zhang, Bob Lew

**Affiliations:** 1Centre for Community Health Studies (ReaCH), Faculty of Health Sciences, Universiti Kebangsaan Malaysia, Kuala Lumpur 50300, Malaysia; caryn@ukm.edu.my (C.M.H.C.); p116885@siswa.ukm.edu.my (Y.K.T.); 2Department of Psychology, University of British Columbia, Vancouver, BC V6T 1Z4, Canada; edklonsky@psych.ubc.ca; 3Australian Institute for Suicide Research and Prevention, World Health Organization Collaborating Centre for Research and Training in Suicide Prevention, School of Applied Psychology, Griffith University, Mount Gravatt, QLD 4222, Australia; boblew@asiacrux.com; 4The Hong Kong Jockey Club Centre for Suicide Research and Prevention, The University of Hong Kong, Hong Kong, China; 5Faculty of Education, Universiti Teknologi MARA, Puncak Alam 42300, Malaysia; fariduddin@uitm.edu.my; 6Centre for Healthy Ageing and Wellness (H-Care), Faculty of Health Sciences, Universiti Kebangsaan Malaysia, Kuala Lumpur 50300, Malaysia; yatieibra@ukm.edu.my; 7Institute of Islam Hadhari, Universiti Kebangsaan Malaysia, Bangi 43600, Malaysia; 8School of Public Health, Shandong University, Jinan 250012, China; jiacunxian@sdu.edu.cn (C.J.); zhangj@buffalostate.edu (J.Z.); 9Department of Sociology, State University of New York Buffalo State University, Buffalo, NY 14222, USA

**Keywords:** three-step theory, suicidal ideation, hopelessness, psychache, college students, China

## Abstract

This study explored the relationship between variables emphasized in the theory’s first step of the three-step theory (3ST)—psychache, hopelessness, and their interaction—to suicide-related variables (i.e., lifetime suicidal ideation and attempt, past-year suicidal ideation, communication of suicidal thoughts, and self-reported future suicide attempt likelihood). Chinese undergraduate students (*N* = 11,399; mean age = 20.69 ± 1.35) from seven provinces participated in this cross-sectional survey. They answered the Suicidal Behaviors Questionnaire-Revised, Psychache Scale, and Beck Hopelessness Scale. Bivariate and multivariate analyses were used to examine the association between psychache, hopelessness, and hopelessness × psychache interaction on the outcome variables. Bivariate analyses showed that psychache and hopelessness were correlated with suicidal ideation and behaviors. In multiple regression models, the interaction between psychache and hopelessness was significantly associated with past-year suicidal ideation and self-report chances of a future suicide attempt, *p* < 0.001, though effect sizes for the interaction term were small. The results are broadly consistent with the 3ST’s proposition of how the combination of pain and hopelessness is related to various suicide-related variables. The low prevalence of suicide-related communication should inform future suicide prevention measures by encouraging help-seeking. Psychache as a correlate of the self-reported likelihood of a future attempt could be further investigated.

## 1. Introduction

Suicide is a leading cause of death among young persons within the 15–29-year-old age bracket [1]. A majority of university students fall within this age range, and they have been found to be at an increased risk of suicidal ideation and suicide attempts compared to the general population [2]. According to the meta-analysis conducted by Mortier et al. [3], the pooled prevalence was 22.3% for lifetime suicidal ideation, 3.2% for lifetime suicide attempts, and 1.2% reported having a suicide attempt in the past year. In China, the pooled prevalence was 10.7% for suicidal ideation [2] and 2.8% for suicide attempts [4]. University students may be at a higher risk of having suicidal ideation and behaviors as they are at the crossroads of adaptation to university life [5]. A meta-analysis further revealed that psychological factors such as the presence of somatization, anxiety, and mental disorders contributed to the highest odds of having suicidal ideation among Chinese university students [6].

Psychological pain (or psychache) and hopelessness have been consistently found to be significant correlates of and risk factors for suicidal ideation and attempts. Psychache is a term that Edwin Shneidman conceptualized and used to describe the psychological pain and emotional turmoil that underlie suicidal behavior [7]. He suggested that psychache is a common stimulus for suicide and identified it as one of the ten commonalities of suicide. A meta-analysis presented that psychache was higher among those who had suicidal ideation and suicide attempts, with greater effect sizes for the former [8]. A longitudinal study has also found that psychache and suicidal ideation were reciprocally associated over time [9]. In China, psychache has been shown to mediate the relationship between psychological strain and suicidal ideation [10].

In the field of suicidology, the concept of hopelessness has its roots in the work of Aaron T. Beck and his colleagues. Beck proposed that hopelessness is a common emotional state associated with suicide. An individual with hopelessness may hold negative views of the self, expect negative outcomes in the future, or believe that one’s current situation will not improve over time [11]. A meta-analysis of longitudinal studies found that higher hopelessness scores predicted a 2.19-fold higher weighted mean odds of having suicidal ideation [12]. Similarly, a psychological autopsy study among Chinese older adults found that hopelessness predicted the highest odds ratio (*OR*: 7.25) for suicide risk among a number of risk factors, including depression [13].

Among the contemporary suicide theories, the three-step theory (3ST) [14] specifies in greater detail the way in which pain and hopelessness combine to lead to suicidal desire. This theory proposes that suicidal desire develops due to the combination of pain (usually psychological) and hopelessness. The theory also states that suicidal desire is more intense when pain exceeds or overwhelms one’s connectedness (to people, valued jobs or roles, or any sense of purpose or meaning) and that intense suicidal desire progresses to action when the dispositional, acquired, and/or practical characteristics of the individual have influenced their capability to make a potentially lethal suicide attempt [14].

The 3ST is relatively new and has been tested in a few contexts (see review by Anderson and Happ [15] and Klonsky et al. [16]), including among university students. In the study by Klonsky and May [14], a model including, pain, hopelessness, and their interaction accounted for 41% of the variance in suicidal desire. This figure was 56% in a large study of UK university students (*N* = 665) [17]. In China, only one extant study, by Yang and colleagues [18], has tested the 3ST. Yang et al.’s study [18] with Chinese college students (*N* = 1097) also found a statistically significant interaction between psychological pain and hopelessness in predicting current suicidal ideation, although the model accounted for only 12% of the variance in suicidal ideation. In addition, one of the study limitations stated was the relatively small number of participants with a suicide attempt history (*N* = 42; 3.8%).

Improved understanding of the development of suicidal ideation is an important first step in suicide prevention. The influence of psychache and hopelessness in a Chinese, non-Western sample could provide local insight into the utility of these suicide risk factors. While the 3ST has been tested in China with moderate support, there have been few large-scale studies that explored the influence of psychache and hopelessness on various outcomes related to suicide risk. Therefore, the aim of this paper was to explore the association between psychache and hopelessness with a spectrum of suicidal thoughts and behaviors (lifetime suicidal ideation and suicide attempt, past-year suicidal ideation, communication of a suicide plan, and likelihood of a future suicide attempt) in a large sample of Chinese university students.

## 2. Materials and Methods

### 2.1. Study Design

This was a cross-sectional survey study of undergraduate students in seven provinces, municipalities, or autonomous regions in China. This design was used to gather data from various undergraduate students and universities in China in order to identify associations between psychache, hopelessness, and suicidality at a specific time point in this population.

### 2.2. Data Collection

We conveniently sampled one university each from Jilin, Qinghai, Shandong and Shaanxi provinces, Ningxia and Xinjiang autonomous regions, and the municipality of Shanghai. In terms of region, Shanghai and Shandong are situated in East China; Shaanxi, Qinghai, Ningxia, and Xinjiang are in Northwest China; and Jilin is located in Northeast China. Each university was chosen using convenience sampling; the universities were associated with some of the authors or their collaborators. The criterion for inclusion was that the university should offer undergraduate programs of study.

Data were collected through pen-and-paper surveys from students enrolled in undergraduate programs from the seven universities. Participant inclusion criteria were being enrolled in an undergraduate program and aged 18 years old or above, while those who were unwilling or unable to provide informed consent were excluded. Quota sampling by year of study was used to ensure that the sample has a fair representation of classes from Year 1 to Year 3 and above, including medical students who take up to five years to complete their undergraduate degree. Undergraduate students were invited to join the study during class by trained research assistants. Participants were first briefed about the aims of the research. They were informed that participation was voluntary. Those who were eligible provided written informed consent. After providing consent, they completed the anonymous questionnaires in approximately half an hour, after which the questionnaires were collected by the research assistant. Participants were given a small gift equivalent to USD 1 as a token of appreciation.

### 2.3. Ethical Considerations

This study received ethical approval from the institutional review board of the School of Public Health, Shandong University (No. 20161103). Hotlines providing counseling services were included in the information sheet according to their availability in each province so that students who experienced any psychological distress during or after the questionnaire administration could contact those services.

### 2.4. Measurements

#### 2.4.1. Demographics

The following demographic data were collected: age, gender, ethnicity, province, and year of study.

#### 2.4.2. Suicidal Behaviors Questionnaire-Revised (SBQ-R)

The SBQ-R, a 4-item questionnaire, was developed to measure a range of suicide-related thoughts and behaviors and was validated in clinical and nonclinical settings [19]. Lifetime suicidal ideation and lifetime suicide attempt were determined by the first item, “Have you ever seriously thought about suicide?”. Participants answer on a range of statements coded 1 = “Never” to 4b = “I have attempted to kill myself and really hoped to die”. A higher score denoted a greater severity of suicidal ideation and attempt. The second item measured the frequency of past-year suicidal ideation, with responses ranging from 1 = “Never” to 5 = “Very often (5 or more times)”. The third item measured communication of a suicide plan, with responses ranging from 1 = “No” to 3b = “Yes, more than once, and really wanted to do it”. Item 4 measured the likelihood of a future suicide attempt, with responses ranging from 0 = “Never” to 6 = “Very likely”. Higher scores on each item denoted a greater severity of suicidal thoughts or behaviors. In a study on suicide risk among China university students, the Cronbach’s alpha coefficient for the scale score was 0.79 [20]. In this study, the Cronbach’s α of the scale score was α = 0.75.

#### 2.4.3. Psychache Scale

The Psychache Scale [21], consisting of 13 items, measures lifetime psychological pain. Items on the scale are scored from 1 = “Never or strongly disagree” to 5 = “Always or strongly agree”. The scale scores have adequate psychometric properties, with internal reliability consistency coefficients > 0.90 when tested among university students. Among Chinese university students, the internal consistency estimate of Cronbach’s α was 0.94 [22]. In this study, the Cronbach’s α of the scale score was α = 0.96.

#### 2.4.4. Beck Hopelessness Scale (BHS)

The Beck Hopelessness Scale, a 20-item questionnaire, measures negative attitudes regarding the short- and long-range future [23]. Participants responded on a 5-point Likert scale, ranging from 1 = “Strongly agree” to 5 = “Strongly disagree”. In a study among China college students, the internal consistency estimate of Cronbach’s α was 0.81 and 0.84 [24]. In this study, the Cronbach’s α of the scale score was α = 0.81.

### 2.5. Data Analysis

Descriptive statistics were used to summarize continuous data, while categorical variables were presented by their frequency and percentage. Normality tests were conducted using the Kolmogorov–Smirnov test and inspection of the histogram and Q–Q plot to ascertain the data distribution of the four items in the SBQ-R. As the data for all items were skewed to the left, Spearman’s rank correlation analyses were used to examine the bivariate relationship between psychache, hopelessness, and the four items on the SBQ-R.

Multiple linear regression analyses were conducted to determine whether psychache (total scale score of the Psychache Scale) and hopelessness (total scale score of the BHS) independently and interactively (hopelessness × psychache) were associated with different suicide-related behavior parameters (item scores of the SBQ-R), controlling for age, gender, and province. To address the issue of multi-collinearity, the predictors were centered (i.e., mean centering).

Cases with missing data were deleted listwise. Diagnostic testing was conducted for outliers, influential points, linearity, and multi-collinearity. A *p*-value of <0.05 was deemed statistically significant for all 2-sided tests. The analysis was conducted using IBM SPSS Statistics (version 27, IBM Corp., Armonk, NY, USA, 2020).

## 3. Results

A total of 11,399 participants (mean age = 20.69 ± 1.35) were included for analysis after excluding missing data (*N* = 1988, 14%) from 13,387 participants who had responded to the questionnaire. The main demographic variables included gender, ethnicity, province, and year of study. Most of the participants were female (*n* = 6987, 61.3%), and the rest were males (*n* = 4412, 38.7%). In terms of ethnicity, a majority were of Han ethnicity (*n* = 8636, 75.8%), while the rest were non-Han (*n* = 2763, 24.2%). About one-third (*n* = 3880, 34.1%) were first-year university students, followed by second-year students (*n* = 3834, 33.7%), and those in their third year and above (*n* = 3667, 32.3%). Participants were well distributed over the three years. In terms of province/autonomous regions, 16.4% (*n* = 1869) were from the Jilin province, while the province with the least participants was Shandong (*n* = 1375, 12.1%). The above information on the composition of the participants reveals a wide range of demographic characteristics. A total of 2274 participants (19.9%) reported past-year suicidal ideation, 3006 (26.4%) reported lifetime suicidal ideation, 226 (2.0%) reported lifetime suicide attempts, 1539 (13.5%) communicated suicide plan, and 655 (5.7%) indicated a self-reported likelihood of a future suicide attempt. The participants’ demographic characteristics and suicidality are presented in Table 1.

Bivariate correlations using Spearman’s rank analysis were determined, which revealed a moderate positive correlation between psychache and hopelessness (ƥ = 0.50, *p* < 0.001). Psychache and hopelessness also correlated significantly with all items on the SBQ-R (Table 2).

Table 3 shows the full model results for the multiple linear regression analyses predicting lifetime ideation and attempt, past-year suicidal ideation, communication of a suicide plan, and self-reported likelihood of a future attempt, adjusting for gender and age. The results show that psychache, hopelessness, and their interaction term could best predict past-year suicidal ideation in a model that accounted for about 14% of the variance in past-year suicidal ideation. The predictive power was weaker for lifetime suicidal ideation and suicide attempts and self-reported likelihood of a future attempt; there was no predictive power at all for communication of a suicide plan (Table 3).

## 4. Discussion

This study aimed to examine the association between psychache, hopelessness, and their interaction with a range of suicide-related variables. The multiple regression models showed that psychache, hopelessness, and their interaction were associated with past-year suicidal ideation, lifetime ideation and attempt, and self-reported likelihood of a future attempt.

The findings extend knowledge about Klonsky’s 3ST model, which states that the combination of psychological pain and hopelessness causes the development of suicidal ideation [14]. While the 3ST proposition is best tested by examining the relationship of pain and hopelessness to current suicidal ideation, or to subsequent suicidal ideation over very short time periods such as hours or minutes (see Klonsky et al. [16] for elaboration), it is also useful to explore how pain and hopelessness relate to other suicide-related outcomes. Results regarding past-year suicidal ideation, i.e., higher hopelessness × psychache was associated with higher past-year ideation, are largely consistent with the 3ST’s proposition that pain and hopelessness motivation the desire for suicide (but not necessarily one’s decision to communicate about that motivation). At the same time, the interactive models in the present paper accounted for much less variance in the suicide-related outcome variables compared to the 41% and 56% variances found in the studies by Klonsky and May [14] and Dhingra et al. [17], respectively. This may be because those studies examined a timeframe consistent with the theory (current pain and hopelessness relating to current suicidal ideation), whereas the present study examined lifetime, past-year, and potential future outcomes.

At the same time, our results are comparable compared to another Chinese study [18] which had reported a 12% variance in current suicidal ideation explained by hopelessness × psychache. The lower variances obtained in these two studies, regardless of past-year or current ideation status, may point to the need to examine developmental and cultural factors that may influence China students differently than the Western samples. For example, a meta-analysis on factors that influenced suicidal ideation among Chinese college students concluded that academic pressure and family relationships were important risk factors for suicidal ideation [6]. Studies have also found that other emotions may trump the influence of hopelessness on suicidal ideation among Chinese students, such as indebtedness [25] and filial piety [26], both of which are related to Confucianism and require further investigation among Chinese undergraduates.

A noteworthy finding of our study was that more than a tenth of respondents (13.5%) indicated that they had communicated a suicide plan at some point in the past year. When taking into account only those who reported lifetime suicidal ideation, 45.3% had communicated their suicide plan to others. This percentage is lower than the proportions of college students who disclosed their suicidal ideation in a study among college students in the US, which indicated a percentage between 53.4% and 53.6% [27], and among Australian adults, at 60.5% [28]. Chinese university students may be less likely to disclose having psychosocial issues probably due, in no small part, to its associated stigma [29], and this is also partially attributable to the Chinese ‘face-saving’ culture [30].

The findings also showed that the interaction between current hopelessness and psychache was associated with self-reported likelihood of future suicide attempts. The participants’ assessment of the likelihood of attempting suicide in the future could be indicative of a current suicidal desire, which makes a future suicide possible. Indeed, previous research has found a relationship of pain and hopelessness to future suicidal ideation or attempts [31]. Even though according to the 3ST, one’s suicidal desire at a given moment is driven by their pain and hopelessness at that moment, pain and hopelessness might have value as a risk factor for future suicidal desire by predicting an increased likelihood of future pain and hopelessness.

The fact that psychache was associated with suicide intention communication should be noted as mental anguish seemed to be associated with individuals seeking help or relief through communicating their suicide intention, and this suggested the need to implement adequate prevention strategies among Chinese undergraduates. We could perhaps target the detection of suicide communication as a psychosocial preventive strategy, considering the fact that about half of suicide decedents communicated suicide intention before death [32,33]. This is because if suicidal intent and tendencies are more effectively understood, those who receive the suicide communication could therefore react with offers of support and help in coping or referral to counselling or psychiatric services.

This study holds significant importance in understanding the relationship between psychache, hopelessness, and suicidal ideation and behaviors among Chinese undergraduate students. The findings suggest that the interaction between psychache and hopelessness was significantly associated with suicidal ideation, which provides support for the first step of the three-step theory (3ST) positing that suicidal desire develops when pain and hopelessness combine. The study results provide actionable insights for counselors or clinicians working with undergraduate students. For example, when a Chinese undergraduate student scores high in psychache and hopelessness, this suggests a heightened risk for current suicidal ideation. In such cases, service providers should assess and monitor the wish to attempt suicide in the future. This study also reveals an essential issue in this context, where only a low proportion of participants who reported suicidal thoughts in the past year disclosed suicide plans. This emphasizes the need for enhanced gatekeeper training among university administrators, educators, and students, which may need to be implemented to more effectively detect any students who may be psychologically distressed and in need of help. Such training can facilitate more effective recognition of psychache and hopelessness among students, enabling timely intervention and support. Help-seeking could also be encouraged by destigmatizing mental health issues and suicidality to potentially reduce the risk of suicide among this population.

This study has a few limitations inherent to a cross-sectional survey design, namely that inferences of causality could not be derived, and self-reported mental conditions could not be verified. The data are not representative of Chinese undergraduate students as one university was chosen from each province. The sampling of universities was carried out using the convenience sampling method. We did not capture the students’ field of study, and not all students enrolled in the sampled universities were made aware of the study availability due to the sampling method. Since we also did not capture the total number of undergraduate students enrolled in the seven universities, the proportion of students who answered the survey could not be estimated. Important cultural aspects, such as the strength of Confucian beliefs and filial piety, were not measured in this study. In addition, we did not explore factors that may lead to the occurrence of psychache and hopelessness, such as the influence of psychological strains. Future studies should take into consideration the quality of important familial relationships and the role of life philosophies in shaping and influencing suicidal ideation and behaviors.

## 5. Conclusions

In conclusion, pain and hopelessness are important risk factors for suicidality among Chinese undergraduates. These findings highlight the need for comprehensive, multipronged approaches that provide support, skills, and resources to alleviate suffering, restore hope, and hopefully reduce suicide risk on Chinese university campuses.

## Figures and Tables

**Table 1 ijerph-21-00885-t001:** Demographic characteristics and suicidality of the participants (*N* = 11,399).

Variable	Frequency (*n*)	Percentage (%)
Age (Mean, SD)	20.69	1.35
Gender		
Male	4412	38.7
Female	6987	61.3
Ethnicity		
Han	8636	75.8
Non-Han	2763	24.2
Province		
Jilin	1869	16.4
Ningxia	1797	15.8
Shaanxi	1775	15.6
Shanghai	1659	14.6
Xinjiang	1538	13.5
Qinghai	1386	12.2
Shandong	1375	12.1
Year of Study		
Year 1	3880	34.1
Year 2	3834	33.7
Year 3 and above	3667	32.3
Past-year suicidal ideation—answered “Yes”	2274	19.9
Lifetime suicide attempt—answered “Yes”	226	2.0
Communicated suicide plan—answered “Yes”	1539	13.5
Likelihood of a future attempt—answered “Likely”, “Rather Likely”, and “Very Likely”	655	5.7

**Table 2 ijerph-21-00885-t002:** Spearman rank correlation of hopelessness, psychache, and items on the Suicidal Behaviors Questionnaire-Revised (SBQ-R).

Variable	Median (IQR)	1	2	3	4	5	6
Psychache (1)	22 (14)	1					
Hopelessness (2)	51 (15)	0.50 ***	1				
SBQ-R 1: Lifetime suicidal ideation/attempt (3)	1 (1)	0.31 ***	0.15 ***	1			
SBQ-R 2: Past-year suicidal ideation (4)	1 (0)	0.33 ***	0.19 ***	0.64 ***	1		
SBQ-R 3: Communication of a suicide plan (5)	1 (0)	0.22 ***	0.11 ***	0.49 ***	0.48 ***	1	
SBQ-R 4: Likelihood of a future attempt (6)	0 (1)	0.27 ***	0.19 ***	0.50 ***	0.47 ***	0.35 ***	1

Note: *N* = 11,399. IQR = interquartile range. *** *p* < 0.001.

**Table 3 ijerph-21-00885-t003:** Linear regression analyses on factors associated with SBQ-R parameters.

Variable	B	Std. Error	Beta	*t*	*p*-Value
SBQ-R 1: Lifetime suicidal ideation/attempt ^a^					
Constant	1.85	0.10		19.01	<0.001
Age	−0.03	0.01	−0.05	−5.38	<0.001
Gender	0.13	0.01	0.09	9.85	<0.001
Province	−0.02	0.00	−0.06	−6.64	<0.001
Hopelessness	0.00	0.00	0.02	2.04	0.041
Psychache	0.02	0.00	0.33	28.54	<0.001
Hopelessness × Psychache	0.00	0.00	−0.04	−4.25	<0.001
SBQ-R 2: Past-year suicidal ideation ^b^					
Constant	1.56	0.10		16.38	<0.001
Age	−0.02	0.00	−0.04	−4.40	<0.001
Gender	0.12	0.01	0.08	9.38	<0.001
Province	−0.02	0.00	−0.05	−5.12	<0.001
Hopelessness	0.01	0.00	0.08	7.55	<0.001
Psychache	0.02	0.00	0.31	27.68	<0.001
Hopelessness × Psychache	0.00	0.00	0.07	6.85	<0.001
SBQ-R 3: Communication of a suicide plan ^c^					
Constant	1.23	0.06		20.64	<0.001
Age	−0.01	0.00	−0.03	−2.69	0.007
Gender	0.06	0.01	0.06	6.92	<0.001
Province	0.00	0.00	−0.01	−0.78	0.437
Hopelessness	0.00	0.00	0.01	0.81	0.420
Psychache	0.01	0.00	0.23	19.80	<0.001
Hopelessness × Psychache	0.00	0.00	−0.01	−0.86	0.390
SBQ-R 4: Likelihood of a future attempt ^d^					
Constant	1.46	0.17		8.78	<0.001
Age	−0.05	0.01	−0.05	−5.82	<0.001
Gender	0.19	0.02	0.08	8.58	<0.001
Province	−0.04	0.01	−0.07	−7.67	<0.001
Hopelessness	0.01	0.00	0.11	10.46	<0.001
Psychache	0.03	0.00	0.25	21.68	<0.001
Hopelessness × Psychache	0.00	0.00	0.04	3.96	<0.001

Note: *N* = 11,399; all predictors are centered; SBQ-R = Suicidal Behaviors Questionnaire-Revised. ^a^
*R*^2^ = 0.11, adjusted *R*^2^ = 0.11, *F* (6, 11,392) = 235.36, *p* < 0.001. ^b^
*R*^2^ = 0.15, adjusted *R*^2^ = 0.15, *F* (6, 11,392) = 323.69, *p* < 0.001. ^c^
*R*^2^ = 0.06, adjusted *R*^2^ = 0.06, *F* (6, 11,392) = 111.54, *p* < 0.001. ^d^
*R*^2^ = 0.11, adjusted *R*^2^ = 0.11, *F* (6, 11,392) = 240.67, *p* < 0.001.

## Data Availability

The data presented in this study are available upon request from the corresponding author.

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
