# Peer review of "Psychache, Hopelessness, and Suicidal Ideation and Behaviors: A Cross-Sectional Study from China"

_ijerph, 2024, doi:10.3390/ijerph21070885_

Round 1

Reviewer 1 Report

Comments and Suggestions for Authors

Author Response

Reviewer 1

Comment #1:

The study gives a clear picture of the association between psychache, hopelessness, and their interaction with a range of suicide-related variables. The study contributes to the literature on the development of suicidal ideation and risk factors.

The reader suggests the following input to improve its publication:

Materials and methodology

Please indicate what the design was used to achieve.

Response from authors #1:

Thank you. This study used a cross-sectional design which is particularly useful for studying this country with a large population. This design allows us to collect information from different undergraduate students or universities simultaneously, providing a snapshot of a population's characteristics, attitudes, or behaviors. By using this approach, we can analyze the association between various variables (between psychache, hopelessness, and their interaction with a range of suicide-related variables) to gain insights. We have now added this to the manuscript as to what this design was used to achieve, as follows:

Lines 106-109: “This design was used to gather data from various undergraduate students and universities in China in order to identify associations between psychache, hopelessness, and suicidality at a specific time point in this population.”

Comment #2:

The study does not indicate how the participants were recruited. Researchers should indicate the recruitment and sampling processes including voluntary participation. The legibility criteria should be clear.

Response from authors #2:

Thank you, we have now rearranged/provided more details of the data collection process, in which the recruitment process including voluntary participation was stated, as follows:

Lines 118-129: “Data were collected through pen-and-paper surveys from students enrolled in undergraduate programs from the seven universities. Participant inclusion criteria were being enrolled in an undergraduate program and aged 18 years old or above, whilst those who were unwilling or unable to provide informed consent were excluded. Quota sampling by year of study was used to ensure that the sample has a fair representation of classes from Year 1 to Year 3 and above, including medical students who take up to five years to complete their undergraduate degree. Undergraduate students were invited to join the study during class by trained research assistants. Participants were first briefed about the aims of the research. They were informed that participation was voluntary. Those who were eligible provided written informed consent. After providing consent, they completed the anonymous questionnaires in approximately half an hour, after which the questionnaires were collected by the research assistant.”

Comment #3:

Ethical approval should be indicated under ethical considerations.

Response from authors #3:

Thank you, we have now added a section “2.3 Ethical Considerations”, as below:

Lines 133-137: “This study received ethical approval from the institutional review board of the School of Public Health, Shandong University (No. 20161103). Hotlines providing counseling services were included in the information sheet according to their availability in each province so that students who experienced any psychological distress during or after the questionnaire administration could contact those services.”

Comment #4:

Results

Table 1 shows the demographic characteristics of the participants. The researchers should highlight the main variables to be clear for the reader, such as gender, ethnicity, etc.

Response from authors #4:

Thank you, we have now provided more information on the main demographic variables, as follows:

Lines 190-199: “The main demographic variables include gender, ethnicity, province, and year of study. Most of the participants were female (n = 6987, 61.3%) and the rest were males (n = 4412, 38.7%). In terms of ethnicity, a majority were of Han ethnicity (n = 8636, 75.8%), while the rest were non-Han (n = 2763, 24.2%). About one-third (n = 3880, 34.1%) were first-year university students, followed by second year (n = 3834, 33.7%) and third year and above (n = 3667, 32.3%). The number of participants were well distributed over the three years. In terms of province/autonomous regions, 16.4% (n = 1869) were from the Jilin province, while the province with the least participants was Shandong (n = 1375, 12.1%). The above information on the composition of the participants reveals a wide range of demographic characteristics.”

Comment #5:

Discussion

The study is very important in examining the association between psychache, hopelessness and their interaction with a range of suicide-related variables. Researchers should indicate the implications of the study.

Response from authors #5:

Thank you. We have now added to the study significance, as follows:

Lines 285-302: “The implications of this study are significant in terms of understanding the relationship between psychache, hopelessness, and suicidal ideation and behaviors among Chinese undergraduate students. The findings suggest that the interaction between psychache and hopelessness was significantly associated with suicidal ideation, which provides support for the first step of the Three-Step-Theory (3ST) positing that suicidal desire develops when pain and hopelessness combine. The findings also highlight the importance of addressing these factors in suicide prevention strategies for counselors or clinicians. For example, when a Chinese undergraduate student scores high in psychache and hopelessness, service providers should assess and monitor for a current suicidal ideation as well as a wish to attempt suicide in the future. Due to the low proportion of participants who communicated their suicide plan despite having past-year suicidal ideation, suicidality may be difficult to detect. However, gatekeeper training may be implemented among school administrators, educators and students to more effectively detect any students who may be psychologically distressed and needing help. Help-seeking could also be encouraged by destigmatizing having mental health issues and suicidality to potentially reduce the risk of suicide among this population.”

Reviewer 2 Report

Comments and Suggestions for Authors

This manuscript presents results of a cross-sectional study of over 11,000 Chinese university students from seven universities in regions across the nation. In a test of Three-Step Theory (3ST), the study measured the levels of psychache and hopelessness among these college students and the relationships with their lifetime suicidal ideation and attempts, as well as suicidal ideation in the past year, their communication of suicidal thoughts, and a self-rating of their likelihood of future suicide attempts. Relationships between psychache and hopelessness were observed as were associations with some of the suicide-related behaviors. General support for the theory was found, supporting the importance of psychological pain and hopelessness for suicidal behaviors. The findings in Chinese students were consistent with those observed in studies of college students in other nations. The authors provide limitations of the investigation and potential cultural issues among Chinese citizens that might influence these issues.

Among the strengths of the study are the sizeable number of participants as well as the inclusion of students from universities from regions across China. Another strength is the testing of the concepts of a theory of suicide that has received minimal attention in studies of Chinese populations. An additional strength is the utilization of instruments with established reliability and validity. The findings underscore the potential influence of culture on behavior, including suicide-related behavior.

There are some issues and questions that might be addressed.

(1) The largest issue of the manuscript lies in the dearth of information about the participants and particularly recruitment. The authors acknowledge in the discussion that the sample is not representativeness with regard to Chinese undergraduate (and some graduate) students with the selection of a single university from each of seven provinces (how many provinces are there in China?). More important is the lack of specificity of how the universities were chosen from each province (information that should have been clearer and included in the respondent section, well before the end of the discussion section). Were these universities associated with the authors? What were the criteria? Additionally, there is no information about recruitment of the 13,000 students (and the 11,000 who provided consent that were included in the data). Were they recruited on each selected campus through online or social media platforms? Through posted or published or distributed calls for participants? Were students recruited from courses in particular fields? Were all those who met the inclusion criteria of age 18+ and undergraduate enrollment on each campus eligible and made aware of the study availability? Was there a target number of participants (e.g., number of men and women and those of each year of university study)? Were the students given class credit or incentives of some kind for their participation? How many students in total were enrolled in each of the seven universities? Without such information, there would be a low possibility that the study could be replicated or to determine representativeness of the study.

(2) The concepts of psychache and hopelessness are central to the reason for the study and the 3ST. However, these concepts are presented almost as if they originated with the developers of 3ST. For instance, the origin of the term and concept of psychache and its meaning derive from the work of Edwin Shneidman (e.g., Definition of Suicide. New York: John Wiley & Sons, 1985). The concept greatly precedes the 3ST. Shneidman suggested that not only psychache (the common stimulus of suicide) but hopelessness-helplessness (the common emotion in suicide) were among the 10 commonalities of suicides he identified. Similarly, the origins of hopelessness in the field of suicidology is most associated with Aaron Beck and others. Some evidence for the role of the concepts central to 3ST are given, but the meaning of the concepts is lacking, where at least a minimal description/definition might be desirable.

(3) Miscellaneous issue – Online 124, the Item 4 measure of likelihood of a future suicide attempt is noted. Though later in the manuscript it is specified that (like the other measures), this is a self-reported likelihood of future suicide attempt, in the case of this particular measure, the inclusion of “self-reported” or some similar designation seems important. This is not a clinical measure of future likelihood of suicide attempt.

Author Response

Reviewer 2

Comment #1:

This manuscript presents results of a cross-sectional study of over 11,000 Chinese university students from seven universities in regions across the nation. In a test of Three-Step Theory (3ST), the study measured the levels of psychache and hopelessness among these college students and the relationships with their lifetime suicidal ideation and attempts, as well as suicidal ideation in the past year, their communication of suicidal thoughts, and a self-rating of their likelihood of future suicide attempts. Relationships between psychache and hopelessness were observed as were associations with some of the suicide-related behaviors. General support for the theory was found, supporting the importance of psychological pain and hopelessness for suicidal behaviors. The findings in Chinese students were consistent with those observed in studies of college students in other nations. The authors provide limitations of the investigation and potential cultural issues among Chinese citizens that might influence these issues.

Among the strengths of the study are the sizeable number of participants as well as the inclusion of students from universities from regions across China. Another strength is the testing of the concepts of a theory of suicide that has received minimal attention in studies of Chinese populations. An additional strength is the utilization of instruments with established reliability and validity. The findings underscore the potential influence of culture on behavior, including suicide-related behavior.

There are some issues and questions that might be addressed.

Response from authors #1:

Thank you for your comments. Please allow us to answer in line to each question raised.

Comment#1a: (1) The largest issue of the manuscript lies in the dearth of information about the participants and particularly recruitment. The authors acknowledge in the discussion that the sample is not representativeness with regard to Chinese undergraduate (and some graduate) students with the selection of a single university from each of seven provinces (how many provinces are there in China?).

Response#1a: We have now added this to the manuscript, through which we hope to provide more information on the coverage and type of region we sampled:

Lines 111-115: “We conveniently sampled one university each from the following: Jilin, Qinghai, Shandong and Shaanxi provinces, Ningxia and Xinjiang autonomous regions, and the municipality of Shanghai. In terms of region, Shanghai and Shandong are situated in East China, Shaanxi, Qinghai, Ningxia, and Xinjiang in Northwest China, and Jilin in Northeast China.”

Comment#1b: More important is the lack of specificity of how the universities were chosen from each province (information that should have been clearer and included in the respondent section, well before the end of the discussion section). Were these universities associated with the authors? What were the criteria?

Response#1b: We have now provided the following information, mainly that the universities were conveniently sampled:

Lines 111-117: “We conveniently sampled one university each from the following: Jilin, Qinghai, Shandong and Shaanxi provinces, Ningxia and Xinjiang autonomous regions, and the municipality of Shanghai. In terms of region, Shanghai and Shandong are situated in East China, Shaanxi, Qinghai, Ningxia, and Xinjiang in Northwest China, and Jilin in Northeast China. Each university was chosen using convenience sampling; the universities were associated with some of the authors or the authors’ collaborators. The criterion for inclusion was that the university should offer undergraduate programs of study.”

Comment#1c: Additionally, there is no information about recruitment of the 13,000 students (and the 11,000 who provided consent that were included in the data). Were they recruited on each selected campus through online or social media platforms? Through posted or published or distributed calls for participants?

Response#1c: We have now provided more information, as follows:

Lines 118-129: “Data were collected through pen-and-paper surveys from students enrolled in undergraduate programs from the seven universities. Participant inclusion criteria were being enrolled in an undergraduate program and aged 18 years old or above, whilst those who were unwilling or unable to provide informed consent were excluded. Quota sampling by year of study was used to ensure that the sample has a fair representation of classes from Year 1 to Year 3 and above, including medical students who take up to five years to complete their undergraduate degree. Undergraduate students were invited to join the study during class by trained research assistants. Participants were first briefed about the aims of the research. They were informed that participation was voluntary. Those who were eligible provided written informed consent. After providing consent, they completed the anonymous questionnaires in approximately half an hour, after which the questionnaires were collected by the research assistant.”

Comment#1d: Were students recruited from courses in particular fields? Were all those who met the inclusion criteria of age 18+ and undergraduate enrollment on each campus eligible and made aware of the study availability? Was there a target number of participants (e.g., number of men and women and those of each year of university study)?

Response#1d: We had used a type of convenience sampling when recruiting the participants, which was quota sampling in order to ensure fair representation of classes from Year 1 to Year 3 and above (lines 122-123). We did not use sex or course type as criteria for the quota sampling. As we had used convenience sampling, the information about the study availability was not communicated university-wide. This is now reflected in the study limitations.

Comment#1e: Were the students given class credit or incentives of some kind for their participation?

Response#1e: Yes, we provided a small incentive, and this is now added to the manuscript, as follows:

Lines 129-130: “Participants were given a small gift equivalent to USD1 as a token of appreciation.”

Comment#1f: How many students in total were enrolled in each of the seven universities?

Response#1f: We did not capture the total number of students who were enrolled in all undergraduate programs in each of the universities we sampled. We acknowledge that these are limitations of the study, which we have now added to the limitations section, as follows:

Lines 306-311: “The sampling of universities was done using the convenience sampling method. We did not capture the students’ field of study, and not all students enrolled in the sampled universities were made aware of the study availability due to the sampling method. Since we also did not capture the total number of undergraduate students enrolled in the seven universities, the proportion of students who answered the survey could not be estimated.”

Comment #2:

(2) The concepts of psychache and hopelessness are central to the reason for the study and the 3ST. However, these concepts are presented almost as if they originated with the developers of 3ST. For instance, the origin of the term and concept of psychache and its meaning derive from the work of Edwin Shneidman (e.g., Definition of Suicide. New York: John Wiley & Sons, 1985). The concept greatly precedes the 3ST. Shneidman suggested that not only psychache (the common stimulus of suicide) but hopelessness-helplessness (the common emotion in suicide) were among the 10 commonalities of suicides he identified. Similarly, the origins of hopelessness in the field of suicidology is most associated with Aaron T. Beck and others. Some evidence for the role of the concepts central to 3ST are given, but the meaning of the concepts is lacking, where at least a minimal description/definition might be desirable.

Response from authors #2:

Thank you for your suggestion. Please find below the additions we have made to the manuscript:

Lines 56-61: “Psychological pain (or psychache) and hopelessness have been consistently found to be significant correlates of and risk factors for suicidal ideation and attempts. Psychache is a term which Edwin Shneidman conceptualized and used to describe the psychological pain and emotional turmoil that underlie suicidal behavior [7]. He suggested that psychache is a common stimulus for suicide and identified it as one of the ten commonalities of suicide.”

Lines 66-70: “In the field of suicidology, the concept of hopelessness has its roots in the work of Aaron T. Beck and his colleagues. Beck proposed that hopelessness is a common emotional state associated with suicide. An individual with hopelessness may hold negative views of the self, expect negative outcomes in the future, or believe that one’s current situation will not improve over time [11].”

[7] Shneidman, E.S. Definition of Suicide. New York: John Wiley & Sons; 1985.

[11] Beck, A.T. Thinking and depression. Arch Gen Psychiatry 1963, 9, 326–333.

Comment #3:

(3) Miscellaneous issue – Online 124, the Item 4 measure of likelihood of a future suicide attempt is noted. Though later in the manuscript it is specified that (like the other measures), this is a self-reported likelihood of future suicide attempt, in the case of this particular measure, the inclusion of “self-reported” or some similar designation seems important. This is not a clinical measure of future likelihood of suicide attempt.

Response from authors #3:

We have now added “self-reported” to “likelihood of a future suicide attempt” where applicable.

Reviewer 3 Report

Comments and Suggestions for Authors

The study has high social and academic relevance. It is well grounded in its theoretical framework, well outlined, and well conducted. The questionnaires are adequate, validated, and quick to answer.

However, a few additional research instruments would make it possible to provide the scientific community with important data on the results of this study (as it is admitted in lines 250 – 257). This would characterize conclusions of low impact in other contexts – which is decidedly not the case of the present research, since there are very few studies on the subject among Chinese university students, thus making the present manuscript valuable for the current knowledge on this problem in this specific context. 

The abstract, however, could be more attractive for the complete reading of the article if, in the end, it replaced lines 36-38 with a synthesis of what appears in lines 232-249 (as these clearly show the social and academic relevance of this work). This would bring the article more readers, citations, and deeper future research.

Author Response

Reviewer 3

Comment #1:

The study has high social and academic relevance. It is well grounded in its theoretical framework, well outlined, and well conducted. The questionnaires are adequate, validated, and quick to answer.

However, a few additional research instruments would make it possible to provide the scientific community with important data on the results of this study (as it is admitted in lines 250 – 257). This would characterize conclusions of low impact in other contexts – which is decidedly not the case of the present research, since there are very few studies on the subject among Chinese university students, thus making the present manuscript valuable for the current knowledge on this problem in this specific context. 

The abstract, however, could be more attractive for the complete reading of the article if, in the end, it replaced lines 36-38 with a synthesis of what appears in lines 232-249 (as these clearly show the social and academic relevance of this work). This would bring the article more readers, citations, and deeper future research.

Response from authors #1:

Thank you for your appreciation of this study. We would like to retain line 36-38 of the abstract. However, we have now added the following to the abstract, as suggested:

Lines 38-40: “The low prevalence of suicide-related communication should inform future suicide prevention measures by encouraging help-seeking. Psychache as a correlate of self-reported likelihood of a future attempt could be further investigated.”